# Bioconversion of a Lignocellulosic Hydrolysate to Single Cell Oil for Biofuel Production in a Cost-Efficient Fermentation Process

Zora S. Rerop †, Nikolaus I. Stellner †, Petra Graban, Martina Haack, Norbert Mehlmer, Mahmoud Masri * and Thomas B. Brück *

Werner Siemens-Chair of Synthetic Biotechnology, School of Natural Sciences, Technical University of Munich, Lichtenbergstraße 4, 85748 Garching bei München, Germany
* Correspondence: mahmoud.masri@tum.de (M.M.); brueck@tum.de (T.B.B.)
† These authors contributed equally to this work.

**Abstract:** *Cutaneotrichosporon oleaginosus* is a highly efficient single cell oil producer, which in addition to hexoses and pentoses can metabolize organic acids. In this study, fed-batch cultivation with consumption-based acetic acid feeding was further developed to integrate the transformation of an industrial paper mill lignocellulosic hydrolysate (LCH) into yeast oil. Employing pentose-rich LCH as a carbon source instead of glucose significantly improved both biomass formation and lipid titer, reaching 55.73 ± 5.20 g/L and 42.1 ± 1.7 g/L (75.5% lipid per biomass), respectively. This hybrid approach of using acetic acid and LCH in one process was further optimized to increase the share of bioavailable carbon from LCH using a combination of consumption-based and continuous feeding. Finally, the techno-economic analysis revealed a 26% cost reduction when using LCH instead of commercial glucose. In summary, we developed a process leading to a holistic approach to valorizing a pentose-rich industrial waste by converting it into oleochemicals.

**Keywords:** fermentation; biotransformations; lipids; single cell oil; biofuel; techno-economic analysis; waste valorization; lignocellulosic hydrolysate; oleaginous yeast





## 1. Introduction

The global energy system is responsible for more than 75% of greenhouse gas (GHG) emissions since it depends primarily on fossil fuels [1]. These emissions accelerate global warming, which impacts our environment, threatens our biodiversity, increases the frequency and intensity of extreme weather events, and damages our economy [2]. In response to these challenges, the UN calls for sustainable and innovative short- and long-term strategies to diversify energy sourcing, accelerate the clean energy transition, and achieve net-zero emissions by 2050 [3,4]. This energy transition demands large-scale deployment of renewable energy sources, such as sustainable advanced biofuels. Furthermore, integrating the up-cycling and end-of-life recycling of urban, agricultural, and industrial wastes into supply chains is a value-adding strategy directed toward the implementation within a truly circular economy. According to the EU's long-term measurement, combining all these approaches can reduce net emissions by around 80% [5]. Such sustainable energy concepts would also reduce dependence on limited resources and disrupted supply chains, the latter of which could be observed during the COVID-19 pandemic. Current biofuel production faces climatic, environmental, and social problems. It relies on globally traded feedstock, such as plant oil or used cooking oil (UCO) [6,7]. The large-scale production of plant oil-based biofuel results in a significant increase in problematic monocultures, increased land use, deforestation, and food competition [6,8–10]. In this respect, biofuel production from domestic biomass waste streams could help lower the external energy dependence

of communities, reduce greenhouse emissions, provide energy storage capability, and implement an efficient route to a circular treatment of waste [8,11].

Oleaginous microorganisms such as the yeast *Cutaneotrichosporon oleaginosus* (ATCC 20509) can utilize these waste streams to produce single cell oil (SCO) as a starting material for advanced biofuel production. As opposed to growing crops for plant oil production, the cultivation of microorganisms is seasonally independent, and biotechnological processes allow for high area efficiency due to vertical scale-up [12,13]. *C. oleaginosus* is one of the most promising oleaginous microorganisms for the production of SCOs as it has been shown to utilize monomers from cellulose, chitin, lignin, and hemicellulose, the most abundant biopolymers on earth [14–16]. The oleaginous yeast displays an excellent ability to grow to very high cell density and to accumulate plant-oil-like lipids with up to 85% of the dry cell weight [8,17]. Moreover, *C. oleaginosus* has a unique ability to simultaneously uptake hexose and pentose sugars, as well as acetic acid, and not follow a typical diauxic cell growth [17]. Furthermore, it shows a high tolerance to inhibitory compounds, such as furans or lignin-derived compounds, such as coumarat and resorcinol, which are major components of depolymerized lignin [15,18,19]. When employed as a high proportion of the carbon source, acetic acid was reported as suitable for *C. oleaginosus*, resulting in high lipid contents of up to 73% but also low biomass generation of only up to 6 g/L [20,21]. Recently, a fermentation process utilizing acetic acid for the cost-efficient production of SCOs with *C. oleaginosus* at high yields was reported [17]. As opposed to other approaches, pure acetic acid was fed to the culture in a consumption-based manner after adding glucose as a starting sugar. To date, this is the process with the highest outcome of SCO, integrating acetic acid as feedstock and using *C. oleaginosus*. Overall, this unconventional yeast is a promising host for the industrial production of SCO as it is efficient, robust, and capable of metabolizing a diverse range of substrates.

The pulp and paper industry is one of the major producers of waste streams, with a high share of biodegradable carbon [22]. The global production of pulp and paper was 188.9 million metric tons in 2020 [23]. Considering that only 30–40% of wood biomass is recovered as cellulose fibers, huge amounts of residual waste are produced. In Europe alone, the pulp and paper industry produces around 11 million tons of waste every year [24]. One of the main processes of pulp production, besides alkaline kraft pulping, is acidic sulfite pulping combined with steam explosion to hydrolyze the plant cells and lignocellulose compounds and separate the valuable cellulose fibers [25]. The pulping waste stream from this process contains high amounts of sugars from hemicellulose (mainly xylose), breakdown compounds from lignin (lignols and lignans), aliphatic carboxylic acids (mainly acetic acid), and furans (mainly furfural and hydroxymethylfurfural) [26,27]. In most industrial plants, the waste stream is heavily concentrated, and the acetic acid therein is evaporated and separated from the rest [28]. To the current date, the released acetic acid is only collected in exceptional cases, such as by the Lenzing AG in Austria [29]. So far, direct combustion for energy production or passing the material into the wastewater is the common strategy to deal with the highly toxic furans, acetic acid, and lignin-derived compounds [30–33]. For these waste streams, specifically for lignin, xylose, and acetic acid, there are no established recovery or valorization systems. However, different lignocellulosic hydrolysates from agriculture residues, such as wheat straw, corn stover, or switch grass, have been applied to produce bioethanol with bacteria or yeasts [34]. Corn steep liquor has been used as a feedstock for lipid production with genetically modified *Rhodosporidium toruloides*, reaching a top-level lipid titer of 39.5 g/L (0.179 g/g yield per sugar) [35]. Moreover, some pilot and demonstration plants have used hard and soft wood chips to produce second-generation bioethanol [36]. Nonetheless, it is important to note that all these examples utilized a lignocellulosic feedstock with a focus on its glucose content, not the pentose sugars or the lignin breakdown products.

For the first time, this study provides an approach for the up-cycling of a crude lignocellulosic hydrolysate, focusing mainly on the valorization of xylose, acetic acid, and lignin-derived compounds. To the best of our knowledge, the fermentative utilization

of a lignocellulosic-rich waste stream directly from a pulping mill and without most of the cellulose and its monomers has never been reported. From this hydrolysate, plant oil-like SCO was produced as a potential domestic feedstock for biofuel or other industrial applications. The current work includes a comprehensive analysis of the hydrolysate, the development of a cost-efficient pretreatment step, and a highly efficient fermentation strategy with the oleaginous yeast *C. oleaginous*. Moreover, the economic viability of the developed process was assessed by a comprehensive techno-economic analysis (TEA).

## 2. Materials and Methods

### 2.1. Analysis of Lignocellulosic Hydrolysate

#### 2.1.1. Sugar Analysis

The sugars and short organic acids were analyzed with high-performance liquid chromatography (HPLC). All the samples were filtered with a 10 kDa filter. The Agilent 1260 Infinity II LC system with Diode Array (DA) and Refractive Index (RI) detectors was used. For separation, a column Rezex ROA-organic H+ 8% from Phenomenex was used with a mobile phase of 5 mM $H_2SO_4$. An isocratic flow of 0.5 mL/min was applied over 60 min with an oven temperature of 70 °C. The detection in the RID was carried out at 40 °C.

#### 2.1.2. Ash

A neutralized and lyophilized lignocellulose hydrolysate of 2 to 4 g was burned at 1000 °C for 3 h to ash. The amount was determined gravimetrically after cooling overnight in a desiccator.

#### 2.1.3. Elemental Analysis

Elemental analysis was carried out with a Euro EA CHNS elemental analyzer (HEKAtech Ltd., Wegberg, Germany). Dynamic spontaneous combustion in a tin boat at approximately 1800 °C was performed with subsequent gas chromatographic separation and was detected using a thermal conductivity detector (TCD).

#### 2.1.4. Quantification of Sulfates

The quantification of sulfates in the hydrolysate was performed chemically with treatment with $CaCO_3$ and $BaCl_2$. The resulting $BaSO_4$ precipitate was quantified gravimetrically.

### 2.2. Strain and Media Composition

#### 2.2.1. Strain

The oleaginous yeast *Cutaneotrichosporon oleaginosus* ATCC 20509 (DSM-11815), supplied by the Deutsche Sammlung von Mikroorganismen und Zellkultur (DMSZ, Braunschweig, Germany), was used for all cultivation and fermentation experiments.

#### 2.2.2. Pretreatment of the Lignocellulosic Hydrolysate

For the precipitation of sulfates within the hydrolysate, 20 g of $CaCO_3$ was added to 1 L of lignocellulosic hydrolysate (LCH) in a 5 L beaker while constantly stirring. The hydrolysate was mixed until the complete outgassing of $CO_2$. Next, the LCH was frozen at −20 °C overnight. After defrosting, the LCH was centrifuged at 16000 rcf for 10 min and the supernatant was collected. Twenty grams of $KH_2PO_4$ was added to the remaining liquid for the removal of excess calcium, and the pH was adjusted with 3 M NaOH to 7. The LCH was centrifuged at 16000 rcf for 10 min and sterile-filtered to be used in the cultivation experiments. All the LCH used for the fermentations was pretreated before its addition to the media.

#### 2.2.3. Pre-Culture

Fifty milliliters of YPD medium (10 g/L yeast extract, 20 g/L peptone, and 20 g/L glucose) in an Erlenmeyer flask containing antibiotics (0.05 g/L kanamycin and 0.1 g/L

ampicillin) was inoculated with a single colony of *C. oleaginosus* (ATCC 20509) from a YPD plate. Antibiotics in fermentation pre-cultures were added to prevent contaminations. The flasks were incubated at 28 °C under constant shaking at 120 rpm for 2 days.

### 2.2.4. Medium Composition for Bioreactor Cultivation

Different media were used for either nitrogen limitation or consumption-based cultivation with acetic acid. The base medium was composed of 0.9 g/L $Na_2HPO_4$, 2.4 g/L $KH_2PO_4$, 4.5 g/L $CH_3COO·Na$, 2 g/L $MgSO_4·7H_2O$, 0.5 $CaCl_2·2H_2O$, 0.00055 mg/L $ZnSO_4·7H_2O$, 0.024 mg/L $MnCl_2·6H_2O$, 0.025 mg/L $CuSO_4 ·5H_2O$, 0.027 mg/L $C_6H_8O_7·Fe·H_3N$, 1 g/L urea, 3 g/L peptone, and 2 g/L yeast extract. For nitrogen limitation, the medium above was adjusted to 1.74 g/L urea, 0.5 g/L yeast extract, and no peptone and $CH_3COO·Na$. The C/N ratio was calculated based on the elemental carbon and nitrogen content, using the information provided by the suppliers of the chemicals. As carbon sources, either glucose, pretreated LCH, or a mix of xylose, glucose, and acetic acid (XGA) was used.

### 2.3. Fermentation
### 2.3.1. Fermentation in 1.3 L Bioreactor

The fermentations were carried out at pH 6.5, at 28 °C, and with a dissolved oxygen content of 50% in a fed-batch process. For the nitrogen-limited and consumption-based acetic acid approaches, a scale of 1 L maximal working volume in the DASGIP® system (Eppendorf AG, Hamburg, Germany) with a total reactor volume of 1.3 L was utilized. In the 500 mL starting medium, the sugar content of 3% from either LCH, glucose, xylose, or XGA was used. The nitrogen content was adjusted to accomplish a C/N ratio of 15 at the start and 65 at the end of the feeding. Pretreated pure and concentrated LCH or XGA with a carbon source content of 260 g/L was continuously fed, starting after 12 h, at a rate of 10 mL/h and 5 mL/h between 36 and 60 h. For non-limited consumption-based (cb) acetic acid feeding, a feed of 90% acetic acid solution was used for the pH regulation. At the same time, the acetic acid functioned as feed for the cultivation. For the combined (co) feeding, the cb-feed was combined with a continuous feed, with the medium mentioned above and a C/N ratio of 16 at the start and a maximum of 37 at the end of the fermentation.

### 2.3.2. Fermentation in 350 mL Bioreactor

The feeding strategy optimization was performed in a maximal working volume of 250 mL in the DasBOX® system (Eppendorf AG, Hamburg, Germany) with a 350 mL total reactor volume. Fifty percent acetic acid was used as consumption-based feed, controlled by the pH change. In the combined feeding condition, LCH was added to 50% (*v/v*) acetic acid to a final concentration of either 10% (*v/v*) and 50% (*v/v*) (50:10 and 50:50 mix of acetic acid:LCH). In the case of continuous feeding, the feeding rates were either 0.5 mL/h or 1 mL/h, starting from 12 h after inoculation, by maintaining a consumption-based acetic acid feed with 50% acetic acid.

### 2.4. Monitoring of Fermentation
### 2.4.1. $OD_{600}$ Measurement

Cell growth was monitored by measuring the optical density at 600 nm ($OD_{600}$). The samples were diluted in the respective cultivation media to an $OD_{600}$ between 0.1 and 1.

### 2.4.2. Dry Weight

The dry weight of the substrate solutions and biomass samples was determined gravimetrically. To measure the dry cell weight, 4 mL of fermentation culture was transferred to pre-weighed reaction tubes, centrifuged (4500 rcf, 20 min), and washed two times with an equal amount of purified water or 50% ethanol in the case of lipid-rich cells. Alternatively, 0.5 mL of a lipid-rich culture was filtered through a pre-weighed 0.2 μm filter paper and washed three times with 2 mL water. The samples were frozen and lyophilized. For each biological replicate, the technical duplicates at least were measured. Growth curve fitting

was performed by the Gompertz function, as shown in Equation (1); here, A is the upper asymptote, $k_G$ is the growth-rate coefficient, and $T_i$ is the time of inflection [37].

$$W(t) = A \times \exp(-\exp(-k_G(t - T_i)))$$ (1)

### 2.4.3. Lipid Content

For the lipid content analysis, the cells from the fermentation were centrifuged and washed two times with 50% ethanol and resolved in water. The cells were disrupted mechanically with a High-Pressure Homogenizer Type HPL6 from Maximator. Triplicates of the 7 mL disrupted cell solution were frozen and lyophilized. The chloroform-methanol lipid extraction was carried out after modifying with the Bligh and Dyer method [38]. Briefly, 100–200 mg biomass was weighed in a glass tube, and 4 mL $Cl_3CH$: methanol (2:1) and and 1 mL $H_2O$ (0.58% NaOH) were added. After 60 min shaking at 120 rpm, it was centrifuged for 10 min at 2000 rcf, and the bottom layer was transferred to a new glass tube. An additional 3 mL $Cl_3CH$: methanol (2:1) was added to the upper phase in the first tube and quickly mixed. After centrifugation, the bottom layer was combined with the lower phase from the first extraction step, and 2 mL of $H_2O$ (0.58% NaOH): methanol (1:1) was added. After mixing and centrifugation, the bottom phase was transferred to a fresh pre-weighed glass tube, and the solvents were evaporated under a nitrogen flow. The lipid amount was determined gravimetrically.

### 2.4.4. Fatty Acid Profile

The fatty acid profile was measured through gas chromatography. Therefore, fatty acid methyl esterification (FAME) of the lyophilized cells was performed; the biomass was not washed, to avoid the loss of very fat cells. Three to ten milligrams of the biomass was weighed in glass vials; all further steps were automated with the Multi-Purpose Sampler MPS robotic from Gerstel. An internal standard of 10 g/L C19 TAG in toluene was used for quantification. First, 490 µL toluene and 10 µL internal standard were added and mixed for 1 min at 1000 rpm, followed by the addition of 1 mL 0.5 M sodium methoxide in methanol, and the solution was heated to 80 °C and shaken at 750 rpm for 20 min. After cooling to 5 °C, 1 mL of 5% HCl in methanol was added, and the mixture was heated to 80 °C while shaking at 800 rpm for 20 min and cooled to 5 °C afterwards. Four hundred microliters of de-ionized $H_2O$ was added and mixed for 30 s at 1000 rpm before 1 mL hexane was added. Extraction was performed by shaking three times for 20 s at 2000 rpm in a quickMix device. The samples were centrifuged for 3 min at 1000 rpm and cooled at 5 °C before a 200 µL sample of the organic phase was transferred to a 1.5 mL vial for chromatography. Gas chromatograph flame ionization detection (GC-FID) was used to quantify the fatty acids. GC-MS was carried out to identify the acids, with the TRACE™ Ultra Gas Chromatograph from Thermo Scientific coupled to a Thermo DSQ™ II mass spectrometer and a Triplus™ Autosampler injector in positive ion mode. A Stabilwax® fused silica capillary column (30 m × 0.25 mm, film thickness 0.25 µm) was used for separation. The temperature profile for the analysis was set to an initial column temperature of 50 °C, increasing at a rate of 4 °C/min up to a final temperature of 250 °C. Hydrogen was used as a carrier gas at a constant flow rate of 35 mL/min. Standardization was performed with the FAMEs Marine Oil Standard (20 components, from C14:0 to C24:1).

### 2.4.5. Statistical Analysis

For statistical analysis of the significant differences in the analysis outcomes, first an ANOVA test was performed. When the ANOVA test showed a test value of $p < 0.001$, and thereby indicated a significant difference, in the next step a Duncan test was applied. The Duncan tests were performed with a test value of $p < 0.01$, and they detected differences and similarities between the test conditions by grouping the results.

*2.5. Techno-Economic Analysis*

To estimate the total capital investment and operation cost for the production of oil from LCH using *C. oleaginosus*, a techno-economic analysis (TEA) was performed. The process was designed with a consequential approach in silico, according to previous laboratory results and available data from the literature [39]. SuperPro Designer (SPD) version 10 (Intelligen, Inc., Scotch Plains, NJ, USA) was used, and the functions describing the biochemical processes were integrated [39]. In the process simulation, a production plant was created with a lipid production capacity of 0.81 metric tons per hour (t/h). The feedstock requirements and the chemicals needed for the LCH pretreatment were estimated based on the media components used in the current work. The mathematical equations and parameters for simulating yeast biomass generation, lipid formation, and enzymatic hydrolysis were based on the results generated in this work or previous experiments [17]. To calculate energy balance, equipment sizing, and purchasing prices, the values from the SPD database were completed with publicly available current prices and data from the literature. The in silico plant featured several operations for LCH pretreatment, fermentation, and downstream processing, including recovery of SCO and recycling of waste streams and side products. The list of modules used in the SPD for the different fermentation conditions (glucose and LCH with acetic acid co-feeding and combination of continuous and consumption-based feeding) is included in the supplements. Lipid productivity was deduced from the lipid titers from the fermentations performed in the 1 L DASGIP® system after 71 h.

$$\text{Lipid productivity (g/L/h)} = \text{lipid titer (g/L)/time (h)} \tag{2}$$

The lipid productivities were calculated for three conditions: glucose with acetic acid co-feeding, LCH with acetic acid co-feeding, and the combination of continuous feed and consumption-based acetic acid co-feeding. The equipment purchase cost (PC) was calculated with an internal SPD function based on the size of the equipment. The installation cost factors relative to the PC were included with values between 1.1 and 1.3 [40]. The cost factors for storage (4%), piping (9%), and site development (5%) were included [41]. The indirect capital costs consisted of insurance (5%), field expenses (5%), construction (10%), contingency (10%), and other expenses (10%). Capital interest was included as 6% of the total investment. Utility and labor costs, as well as waste disposal, were estimated based on the current prices in Germany [42]. The costs for consumables and raw materials were set according to current market prices. Maintenance and insurance were included as faculty-dependent costs at 3% and 0.7% [41].

## 3. Results

*3.1. Analysis of the Lignocellulosic Hydrolysate*

The lignocellulosic hydrolysate (LCH), coming from the spent liquor of the pulping process, was comprehensively analyzed to gain a better understanding of its composition and to set up a pretreatment strategy. The pH was 1.7, with a dry mass of 247.7 ± 13.9 g/L and a high sulfate content of 19.4 ± 2.0 g/L. HPLC analysis was used to quantify the sugar, organic acids, and furans. It showed that the primary sugar of the hydrolysate was xylose, with 77.05 ± 2.46 g/L, and smaller amounts of other sugars, such as glucose (11.51 ± 0.39 g/L), mannose (8.21 ± 0.84 g/L), galactose (6.47 ± 0.21 g/L), and others (12.37 ± 1.10 g/L), resulting in a total sugar content of 115.60 ± 4.99 g/L. The organic acids had a total concentration of 16.83 ± 1.43 g/L, acetic acid being the major one with 12.34 ± 1.20 g/L. A considerable amount of hydroxymethylfurfural (HMF), with 4.53 ± 0.59 g/L, and furfural, with 0.68 ± 0.04 g/L, was detected. The hydrolysate is a waste stream from the industrial production of cellulose fibers using hardwood as a source material in an acidic sulfite pulping process. The chemical hydrolysis of lignocellulose results in the detected sugar monomers, organic acids, and breakdown compounds from lignin. The total ash was 0.70 ± 0.01 g/L, with low concentrations of phosphorus (0.035 g/L) and no detectable nitrogen. The rest of the material derives from lignols and

lignans as well as other plant metabolites, resulting in a total amount of $89.95 \pm 22.97$ g/L. Table S1 gives a summarized overview of the composition.

### 3.2. Pretreatment of the Lignocellulosic Hydrolysate

To transform the waste stream LCH into a suitable carbon source for yeast fermentation, the challenges of low pH and the lack of essential nutrients needed to be addressed with a suitable pretreatment strategy. However, the formation of insoluble particles was observed after the addition of the other medium compounds such as buffering salts, vitamins, nitrogen, and trace elements, as well as during the cultivation. To address this problem, some simple pretreatment steps were taken. As a result of the chemical hydrolysis from the industrial pulping process, the LCH has a high sulfate content and a highly acidic pH. To reduce the amount of residual sulfate, overliming was performed using $CaCO_3$. Nevertheless, particle formation could still be observed. Titration of the overlimed LCH with $H_2PO_4^-$ or $HPO_4^{2-}$ salts was empirically found to prevent this particle formation. As this process still results in an acidic pH, a final neutralization with NaOH was performed after the $KH_2PO_4$ treatment. The details of the different tested pretreatment conditions are provided in the Table S2. The reduction in the overall bioavailable carbon by the pretreatment ($159.85 \pm 2.80$ g/L) was negligible compared to the starting material ($162.05 \pm 1.44$ g/L). This qualifies the LCH pretreatment as a suitable step in the preparation of the waste stream as a carbon source for bioreactor fermentations. This optimized pretreatment strategy was applied to all the LCH used for fermentation in this study.

### 3.3. Nitrogen-Limited Fermentation

Fermentation under nutrient limitation, such as with phosphate or nitrogen, is a common strategy to trigger lipid accumulation in oleaginous yeast [43]. Therefore, nitrogen-limited fermentation was performed using LCH as a carbon source (Figure 1a). To understand the effect of the lignocellulose-derived compound on the yeast metabolism, a further control medium was composed. The control medium contained only the main sugars from the LCH in their respective ratios: xylose (8.28%), glucose (1.03%), and acetic acid (0.69%), and it is therefore abbreviated with xylose, glucose, and acetic acid (XGA). The fermentations resulted in a biomass formation of $7.02 \pm 0.88$ g/L and $16.65 \pm 0.24$ g/L for LCH and XGA, respectively, showing better growth performance on the XGA medium.

### 3.4. Acetic Acid-Based Fermentation with Commercial Sugars as Carbon Source

The successful production of SCO from refined sugars with acetic acid-based fermentation was shown previously [17]. In this work, LCH was integrated as a carbon source for fed-batch fermentation with consumption-based acetic acid feeding. As controls, glucose and xylose, as well as the XGA mix, as starting sugars were tested. The analysis showed that the concentration of acetic acid remained constant due to the consumption-based feeding; the sugars were consumed within the first 40 h after inoculation, as shown in Figure 1b–d. The biomass increased further after the full consumption of the sugar and reached $35.11 \pm 1.11$ g/L, $39.95 \pm 4.59$ g/L, and $39.00 \pm 0.76$ g/L for glucose, xylose, and XGA by 71 h, with lipid titers of $18.5 \pm 3.6$ g/L, $23.6 \pm 0.5$ g/L, and $21.6 \pm 2.8$ g/L for glucose, xylose, and XGA, respectively (Figure 2a). The same applied to the carbon-to-lipid conversion yield of lipid carbon from substrate carbon, which was $0.204 \pm 0.003$ g/g, $0.230 \pm 0.001$ g/g, and $0.235 \pm 0.001$ g/g for glucose, xylose, and XGA, respectively (Figure 2b). The fatty acid profile of the oil is shown in Figure 2c; the most abundant fatty acid was C18:1 with around 54%, followed by C16:0 (~24%), 18:0 (~15%), and C18:2 (~6%) and by traces of C18:3, C16:1, and C22:0 with less than 0.3% each.

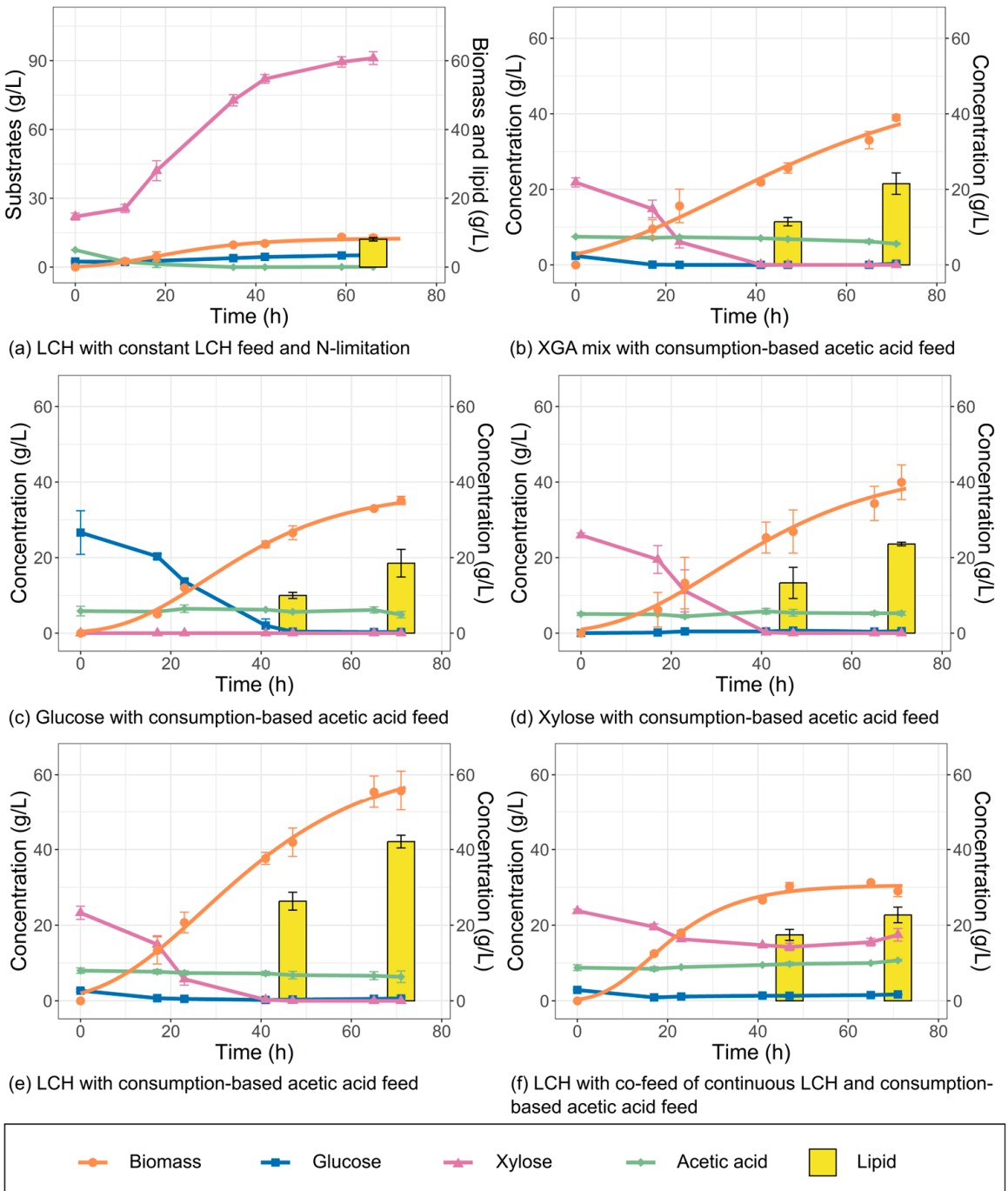

**Figure 1.** Biomass, substrate—namely glucose, xylose, and acetic acid—concentration, and lipid titer for different fermentation conditions. (**a**) LCH with constant feed and N-limitation; (**b**) XGA mix with consumption-based acetic acid feed; (**c**) Glucose with consumption-based acetic acid feed; (**d**) Xylose with consumption-based acetic acid feed; (**e**) LCH with consumption-based acetic acid feed; (**f**) LCH with co-feed of continuous LCH and consumption-based acetic acid feed. Fermentations were performed in triplicates (except for glucose and xylose, which were performed in duplicates). Error bars display two times the standard deviation. LCH—lignocellulosic hydrolysate, XGA—xylose, glucose, and acetic acid mix as model substrate.

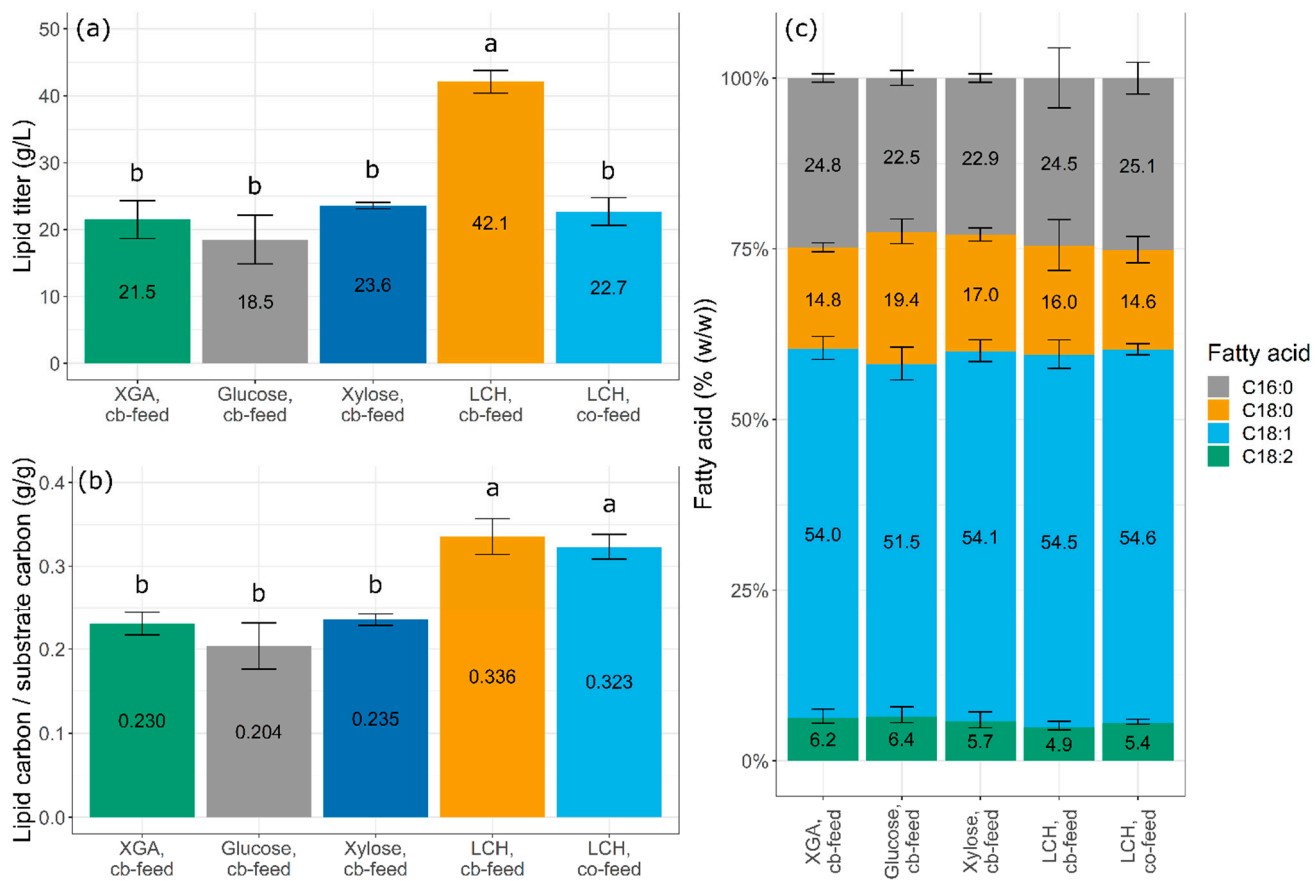

**Figure 2.** Lipid analysis of the four most important fermentation conditions. (**a**) Lipid titers after 71 h fermentation in 1 L scale. (**b**) Carbon conversion yield from substrate carbon to lipid carbon. (**c**) fatty acid profile of the main fatty acids quantified with GC-FID. Error bars display two times standard deviation; Duncan tests were performed with $p < 0.01$ to determine statistical groups a and b for lipid titer and carbon conversion, Duncan results for the fatty acid profile are shown in the Table S3. cb-feed—consumption-based acetic acid feed; LCH—lignocellulosic hydrolysate; LCH co-feed—lignocellulosic hydrolysate as starting carbon, and constant feed with cb-feed.

### 3.5. Acetic Acid-Based Fermentation of Lignocellulosic Hydrolysate at Different Starting Concentrations

The LCH used in this study is the spent liquor from acidic hardwood pulping, containing sugars, mainly xylose, as well as acetic acid. LCH further contains furans and lignin breakdown products that can have an inhibitory effect on microorganisms. To assess the optimal balance between carbon supply and the toxic level of inhibitory compounds, fermentations with three different starting concentrations of LCH were performed. The amount of hydrolysate corresponded to the concentrations of 3%, 5%, and 7% of the bioavailable carbon in the starting medium. The fermentation with the highest LCH concentration of 7% showed a slower overall growth with a growth-rate coefficient of 1.21 compared with 1.33 for 3%. The dry biomass formations were 39.64 ± 6.30 g/L and 51.92 ± 0.18 g/L for 7% and 5%, compared to 55.73 ± 5.20 g/L in the case of 3%. Accordingly, an increase in the concentration to 5% resulted in a slightly longer lag phase and a slightly lower biomass accumulation after three days of fermentation. The lipid titer was lower for 7% LCH (19.83 ± 1.38 g/L) and 5% (25.75 ± 2.02 g/L) and much higher for 3% (42.1 ± 1.7 g/L). The starting carbon source concentration of 3% resulted in the shortest lag phase, the best growth rate, and the overall biomass formation, as shown in Figure S1.

### 3.6. Acetic Acid-Based Fermentation on Lignocellulosic Hydrolysate with Optimized Conditions

The optimized conditions for the fed-batch fermentation of *C. oleaginosus* on LCH were carried out in three biological replicates. The starting carbon content of the hydrolysate corresponding to 3% sugar and acetic acid was used. A high biomass formation of $55.73 \pm 5.20$ g/L was achieved after 71 h, which is a seven-fold increase compared to the nitrogen-limited fermentation of LCH. Moreover, the biomass formation was about 50% higher with LCH as a carbon source compared to glucose or xylose under equivalent reaction conditions (Figure 1b,c). However, the sugar consumption showed a similar profile to that of the fermentation processes using sole sugars. The lipid titer after 71 h was, with $42.1 \pm 1.7$ g/L (lipid productivity 0.593 g/L/h), twice as high as that from the glucose fermentation ($18.5 \pm 3.6$ g/L, productivity 0.261 g/L/h) (Figure 2a). The carbon conversion was 0.336 g/g lipid carbon from substrate carbon (Figure 2b) with $76.05 \pm 9.04\%$ lipid per dry biomass. The fatty acid profile did not show significant changes compared to the lipid profile of the fermentation on xylose or XGA and slight changes compared to the fermentation on glucose (Figure 2c and Table S3).

Furthermore, lignin breakdown products were shown to be metabolized by *C. oleaginosus*, in a range of 91.6% of the initial compound amount (Figure S2 and Table S4).

### 3.7. Screening of Different Strategies for Feeding Lignocellulosic Hydrolysate

In several fermentation approaches, the combined feeding of a mix of LCH and acetic were tested. The biomass of the cells fed with a 50:10 (50% acetic acid, 10% LCH, 40% purified water) and a 50:50 (50% acetic acid, 50% LCH) mix of acetic acid:LCH were $51.5 \pm 3.7$ g/L and $59.6 \pm 1.2$ g/L. Those solely fed with acetic acid after 65 h ($50.8 \pm 0.1$ g/L), with 50:50 co-feeding, resulted in the highest biomass (Figure 3). The lipid titers after 65 h were $28.4 \pm 0.4$ g/L and $30.4 \pm 1.4$ g/L for the 50:10 and 50:50 feeding, respectively, while $25.2 \pm 3.1$ g/L was measured for the control settings (Figure 3). As an alternative feeding strategy, the pure pretreated LCH was continuously fed into the reactor throughout the fermentation process at two feeding rates: 0.5 mL/h or 1 mL/h, in addition to the consumption-based feeding of acetic acid. The dry biomass accumulated after 65 h was $50.4 \pm 2.2$ g/L and $50.4 \pm 4.6$ g/L for 0.5 mL/h and 1 mL/h, respectively. The maximum lipid titers were $0.5 \pm 2.4$ g/L for 0.5 mL/h and $26.3 \pm 2.7$ g/L for 1 mL/h and were therefore above the control ($25.2 \pm 3.1$ g/L). The control and the two best feeding strategies are visualized in Figure 3; a comparison of all five strategies can be found in the Figure S3.

To select the setup with the highest LCH turnover, the share of LCH in the overall consumed carbon was calculated. The strategy using 1 mL/h of continuous feeding resulted in the highest percentage of LCH consumed ($21.3 \pm 2.9\%$), followed by 50:50 acetic acid:LCH (12.5%) and continuous feed with 0.5 mL/h ($10.4 \pm 1.2\%$). For the differences between the duplicates continuously fed with LCH results and the different consumed quantities of acetic acid, see Table S6. As the share of LCH using 50:50 acetic acid:LCH stayed constant over the fermentation, no standard deviation could be calculated. In addition, the carbon conversion rate of fed carbon to lipid was calculated for all conditions. In comparison, the setups using continuous feed had the highest carbon conversion yield, with $0.198 \pm 0.002$ g/g for 0.5 mL/h and $0.188 \pm 0.0002$ g/g for 1 mL/h. The control showed a conversion yield of $0.164 \pm 0.0002$ g/g, and co-feeding with the acetic acid:LCH mix resulted in $0.167 \pm 0.0001$ g/g, for a ratio of 50:10 and $0.167 \pm 0.0005$ g/g for 50:50.

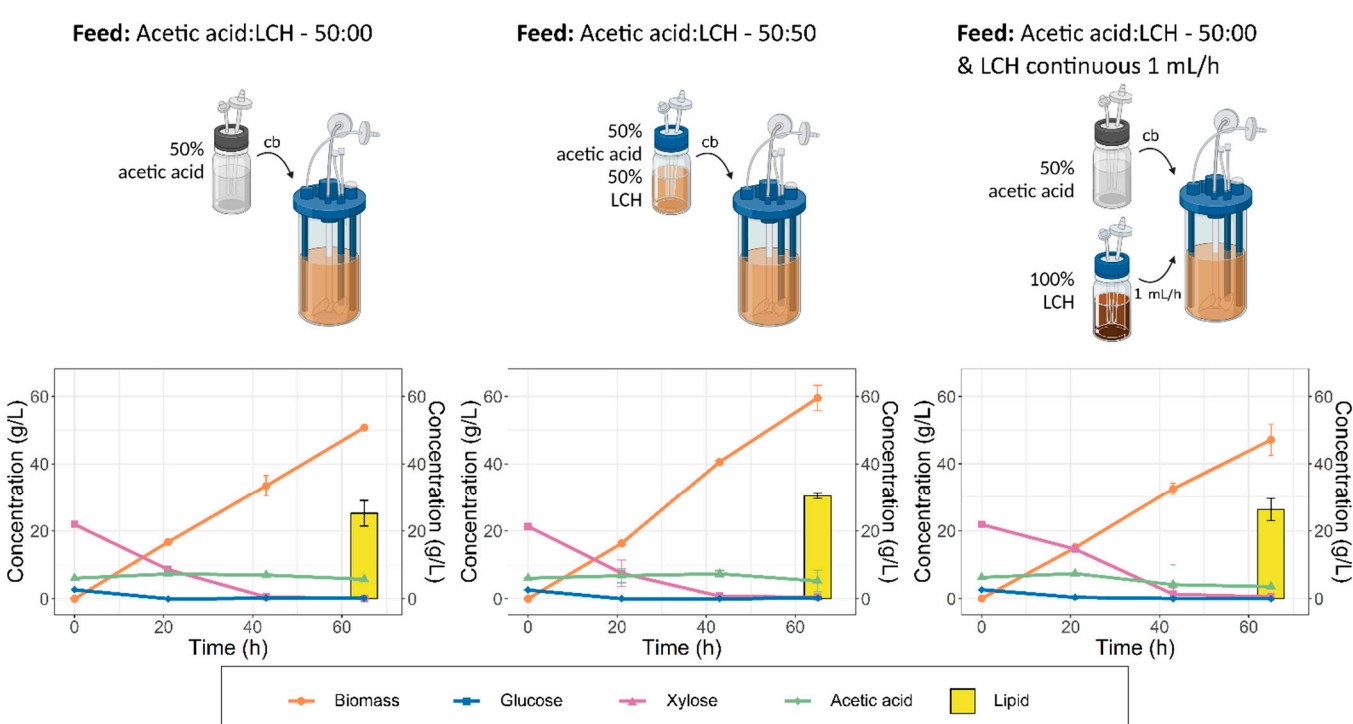

**Figure 3.** Comparison of the different feeding strategies in 0.25 L scale in the DASbox® system. Below each schema, the biomass accumulation, substrate concentration, and total lipid titer after 65 h are shown. On the left side, the consumption-based acetic acid feed with a 50% acetic acid solution is displayed (acetic acid:LCH 50:00), in the middle the consumption-based feed mixture of LCH and acetic acid (acetic acid:LCH—50:50), and on the right side the combination of continuous feeding with LCH at 1 mL/h and consumption-based acetic acid feeding (acetic acid:LCH 50:00 and LCH continuous). Fermentations were performed in duplicates at least. Error bars display two times standard deviation. LCH—Lignocellulosic hydrolysate, cb—consumption-based acetic acid feed.

### 3.8. Fermentation with Co-Feeding of Acetic Acid and Lignocellulosic Hydrolysate

Using 1 mL/h continuous feed of LCH (LCH co-feed) led to the highest share of LCH consumed of the overall carbon source and had the second highest carbon conversion yield among all the tested conditions. Therefore, this condition was selected for the 1 L DASGIP® system. The slight scale-up also allowed a better direct comparison of the fermentation output with the other conditions tested in the DASGIP® system. A consumption-based feed with 90% acetic acid was used in combination with a continuous LCH feed of 3.3 mL/h. The biomass accumulation reached $29.0 \pm 5.87$ g/L after 71 h, with a lipid titer of $22.7 \pm 1.95$ g/L (lipid productivity 0.320 g/L/h). HPLC analysis showed that the sugars and acetic acid were evenly consumed throughout the fermentation, and only xylose accumulated slightly from $15.5 \pm 1.11$ g/L at 65 h to $17.44 \pm 1.85$ g/L at 71 h, as shown in Figure 1f. The xylose accumulation could presumably be solved by using a slightly lower feeding rate, resulting in a steady supply of sugar and fewer inhibitory compounds. When compared to the other conditions tested in the DASGIP® system, biomass accumulation with LCH co-feed was only approximately 17% less than that with glucose as the starting sugar. The lipid titers were comparable to the titers reached with glucose or xylose, with the LCH co-feed performing only 4% worse than the titers reached with xylose.

### 3.9. Techno-Economic Analysis

Aside from the experimental results in this study, a consequential TEA was performed to assess the economic feasibility of a commercial-scale production plant using LCH as an industrially relevant feedstock. Two different modes of LCH fermentation from the 1 L

scale DASGIP® experiments were implemented and compared to the setup with glucose as a carbon source, as shown in Figure S4. Firstly, there was the consumption-based acetic acid feeding using LCH as a starting carbon source, and secondly, there was the combination of consumption-based acetic acid and the continuous feeding of LCH. Both fermentation modes were compared to an operation setup, where glucose was used as the starting carbon source for consumption-based acetic acid feeding. According to the simulation, the capital expenditure (CAPEX) was lowest for the LCH cb-feed at USD 30.1 M. In comparison, the CAPEX in the glucose cb-feed was USD 42.5 M and USD 41.8 M for the LCH co-feed. The increased capital cost was mainly caused by the different amounts and the vessel volumes of the fermentation vessels in the respective model. Different vessel combinations had to be used for the different fermentation strategies because of differences in the dwell times to simulate comparable lipid productivities. Other than the vessel amounts and the volumes, the equipment costs were the same. Depreciation was calculated as 10% over ten years, resulting in USD 4.2 M, USD 3.0 M, and USD 4.2 M for glucose cb-feed, LCH cb-feed, and LCH co-feed, respectively. The quantity and volumes of the vessels also resulted in differences in the operating costs, especially the costs for power (glucose cb-feed: USD 4.8 M, LCH cb-feed: USD 2.3 M, LCH co-feed: USD 4.0 M), cooling water (glucose cb-feed: USD 1.0 M, LCH cb-feed: USD 0.5 M, LCH co-feed: USD 0.9 M), labor cost (glucose cb-feed: USD 2.2 M, LCH cb-feed: USD 1.5 M, LCH co-feed: USD 1.8 M), maintenance (glucose cb-feed: USD 0.8 M, LCH cb-feed: USD 0.6 M, LCH co-feed: USD 0.8 M). Raw materials were a major expense, dominated by the acetic acid price. For the main analysis, an acetic acid price of 600 USD/t was used, resulting in a raw material cost of USD 8.5 M for glucose cb-feed, USD 7.7 M for LCH cb-feed, and USD 7.2 for LCH co-feed. The final costs for the yeast oil calculated from the in silico model were 3985 USD/t (glucose cb-feed), 2938 USD/t (LCH cb-feed), and 3576 USD/t (LCH co-feed), respectively (Figure 4a).

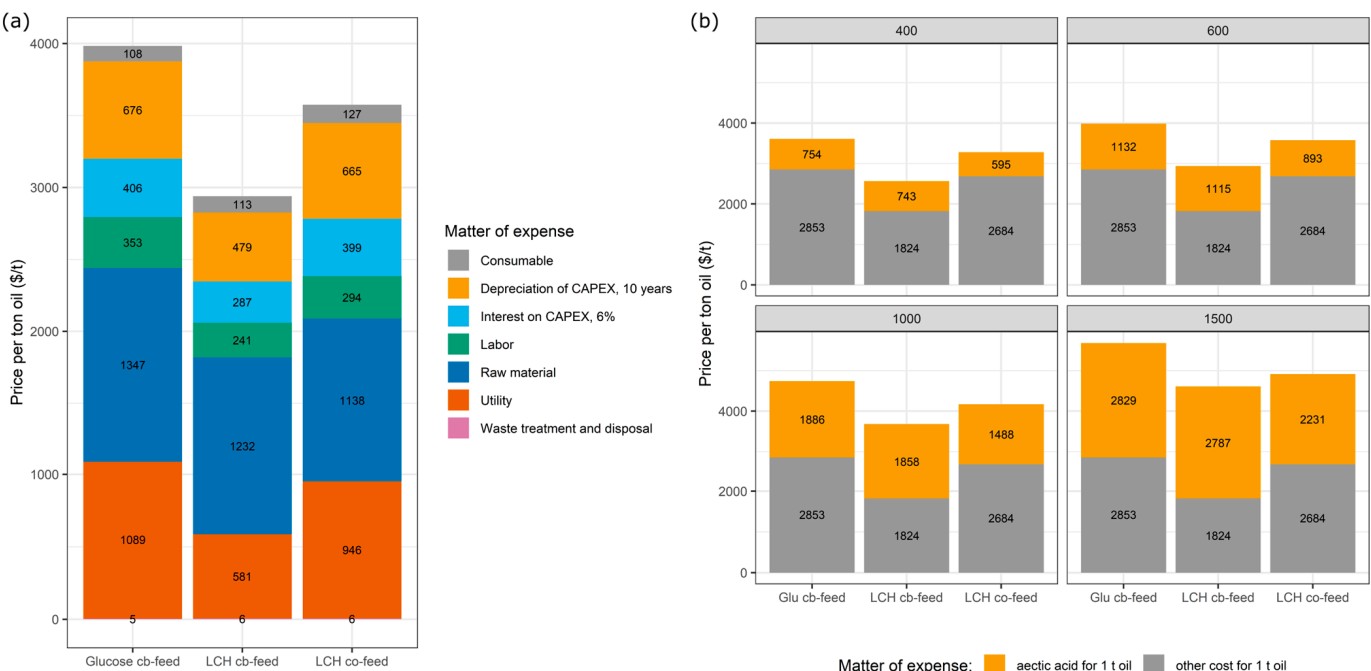

**Figure 4.** (**a**) Annual production cost of yeast oil in USD/Mt for the three fermentation strategies analyzed with the TEA. The respective amount of feedstock was set to produce yeast oil at a rate of 0.81 Mt/h (LCH cb-feed: 1 Mt/h, glucose cb-feed: 0.151 Mt/h, LCH co-feed: 2.1 Mt/h). (**b**) Annual production costs in a sensitivity analysis of the influence of the acetic acid prices 400, 600, 1000, and 1500 USD/t. LCH—lignocellulosic hydrolysate; cb-feed—consumption-based acetic acid feed; co-feed—lignocellulosic hydrolysate as starting carbon and constant feed with cb-feed.

In addition, a sensitivity analysis was performed considering different acetic acid prices, as it accounted for the major part of the raw material costs (glucose cb-feed: 84%, LCH cb-feed: 90%, LCH co-feed: 78%). Acetic acid prices in Western Europe fluctuated between 400 USD/t and 1500 USD/t between 2012 and 2023; so, the cost of yeast oil per t is largely dependent on the current acetic acid price [44,45]. Therefore, acetic acid costs of 400 USD/t, 600 USD/t, 1000 USD/t, and 1500 USD/t were assumed for the sensitivity analysis. The scenario using LCH cb-feed responded strongest to the price changes, as shown in Figure 4b. The price decrease to 400 USD/t led to a 12.6% decrease in the cost to 2567 USD/t oil. The rise to 1000 USD/t led to a 25.3% increase in production costs (3682 USD/t), and an elevated acetic acid price of 1500 USD/t resulted in a 56.9% increase (4611 USD/t). These tendencies were similar for the glucose cb-feed and LCH co-feed. Overall though, LCH cb-feed still showed the cheapest production costs for a decrease or increase in acetic acid prices.

## 4. Discussion

The lignocellulosic hydrolysate from acidic hardwood pulping mainly consisted of xylose, glucose, and other sugars, as well as acetic acid and lignin breakdown products, as expected and described by the literature [26,27]. This composition makes LCH an ideal carbon source for *C. oleaginosus,* which has been shown to efficiently metabolize both hexose and pentose sugars as well as acetic acid [17]. Waste streams with a high content of xylose and lignin-derived compounds are of special research interest because only a few commercial valorization strategies are available [46,47].

Due to the low amounts of nitrogen, phosphorus, and other elements, such as sulfur, magnesium, or calcium, additional nutrient supplementation is required before using the hydrolysate in a fermentation medium. Without pretreatment the LCH formed insoluble particles when combined with the other media components, which was presumably the result of sulfate and phosphate salt formation.

The pretreatment strategy established in this study produced a neutral LCH that did not form insoluble material when buffering salts or trace elements were added. As the sugar content was maintained, it could be efficiently applied as a carbon source for *C. oleaginosus*.

Primarily, the conventional nitrogen-limited fermentation approach was applied for the comparison of the LCH-containing medium with the XGA model. The reduced growth in LCH conditions was presumably caused by the high concentration of inhibitory furans and phenolic compounds in the medium. However, the contained phenolic and other lignin-derived compounds, did not inhibit growth completely. Furthermore, *C. oleaginosus* has already been shown before to metabolize some of the lignin breakdown compounds [16]. Generally, nitrogen limitation induces lipid formation but may restrict biomass formation at the same time, as nitrogen is required for protein synthesis and other metabolic processes [48]. Due to these drawbacks, nutrient limitation strategies are difficult for the production of SCO with the high yields required for commercial production.

The successful production of SCO with *C. oleaginosus* from refined sugars with acetic acid-based fermentation was shown previously. This approach combines a non-limiting sugar-containing starting medium with a consumption-based acetic acid feeding strategy [17]. For a commercially more attractive process, the LCH as a waste product was applied as a starting carbon source in this study, in place of costly sugars. This was further advanced in the fed-batch fermentations of this study. Xylose and glucose as sole carbon sources showed a similar growth behavior to the fermentation using XGA, which indicates a good and simultaneous uptake of both glucose and xylose. This is a considerable advantage of *C. oleaginosus* as a fermentation strain because the full potential of the biomass as a carbon source can be exploited. Moreover, the acetic acid can theoretically be directly channeled into the metabolism of lipid synthesis by acetyl-CoA synthetase. Therefore, a more efficient method of lipid biosynthesis in *C. oleaginosus*, compared to nutrient limitation conditions, can be employed [17]. Fortunately, acetic acid is part of the pulp and paper

waste streams and hence represents an industrially more promising carbon source than refined sugars, as it is a byproduct of hardwood pulping alongside the LCH [29,49].

The fatty acid profile of the SCO produced showed a typical composition of oleaginous yeasts, mainly consisting of C16 to C18 fatty acids, which is favorable for biodiesels [50]. Bio-based diesel from plant oils is used in the form of fatty acid esters. One main factor in evaluating the suitability of the fatty acids for biodiesel is the cetane number. This number indicates the combustion speed; it increases with chain length and decreases with saturation. The minimal overall values for use as biodiesel are 47 (U.S. American specification) and 51 (European specification). Examples of cetane ranges of biodiesels are 48–67 (soy oil) and 60–63 (palm oil) [51,52]. Among the unsaturated fatty acids, oleic acid has the greatest cetane number of 56 and is most prominent in the fatty acid profile of *C. oleaginosus* [53]. The cetane numbers of the other dominating fatty acids are 75 (palmitic acid) and 76 (stearic acid) [53]. As C16:0, C18:0, and C18:1 compose around 93% of the fatty acids, based on its composition the oil of *C. oleaginosus* fits well as a raw material for biodiesel.

The starting carbon source concentration of 3% resulted in the shortest lag phase, the best growth rate, and the overall biomass formation, demonstrating that the combination of monomeric sugars and acetic acid in the LCH can be metabolized efficiently and simultaneously by *C. oleaginosus*. Higher concentrations resulted in higher concentrations of inhibitory furans and lignin breakdown products, explaining the longer lag phase and decreased growth rates. Therefore, the starting carbon concentration of 3% was used for all further experiments.

The optimized fermentation strategy of *C. oleaginosus* on LCH with acetic acid consumption-based feeding showed the best performance in biomass accumulation, lipid titer, and lipid yield, with minor changes in the lipid profile. The increased performance of LCH as a substrate might be attributed to the contained lignol and lignan compounds that can partly be metabolized by *C. oleaginosus* [16]. In summary, this combination of the waste stream and oleaginous yeast is promising for a sustainable new process of SCO production. It combines the application of a waste stream rich in xylose, lignols, and lignans with a fermentation strategy resulting in high lipid yields. The lipid titer of 42.1 ± 1.7 g/L with a wild-type *C. oleaginosus* from this study is higher than the top yields from the literature of 39.5 ± 0.49 g/L with a genetically modified *R. toruloides* [35]. In the measured lipid carbon conversion, both sugars and acetic acid were considered but not the lignols and lignans from the hydrolysate. Nevertheless, the conversion yield of 0.336 g/g per carbon source is close to the theoretical maximum for xylose of 0.34 g/g [54]. Furthermore, efficient metabolization of the lignin breakdown products was shown, which can explain the high conversion yield and demonstrates the potential of *C. oleaginosus* for the valorization of waste streams from the pulp and paper industry.

Consumption-based acetic acid feeding results in a high lipid productivity of over 80%, as shown in the literature [17]. However, compared to acetic acid or commercially available sugars, LCH is a cheap feedstock as it is a waste product with limited options for value creation. To increase LCH uptake by the oleaginous yeast relative to the consumption of acetic acid and to better balance the two waste streams, different fermentation modes were compared. For the overall evaluation of the feeding strategies, the focus was set on the share of the consumed LCH and the lipid titer. With regard to this, the two best feeding strategies were cb-feed with 50:50 acetic acid:LCH as well as a co-feed with a continuous feed of 1 mL/h LCH. Providing more LCH during the fermentation process, these strategies could significantly increase the share of LCH. At the same time, the starting concentration of inhibitory compounds was not increased, which prevents an elongated lag phase, as observed for higher LCH concentrations.

The consequential TEA revealed a price reduction of 26% using LCH instead of glucose. Looking forward, the price points for the LCH processes could potentially be reduced by the further optimization of the fermentation processes and the resulting lipid conversion yields. All the calculated price regimes are in the current price range for organic palm oil (2500–3000 USD/t) [55]. Nevertheless, the consequential TEA, as applied in this study,

focuses mainly on the changes in the input and output conditions relative to each other [56]. The calculated prices per t should therefore be considered as an estimate of magnitude. Other plant oils used for biofuel production, such as canola oil, are priced well below 1000 USD/t [57]. However, utilizing an industrial waste stream circumvents the competition for edible oils, thus enhancing the sustainability aspect of the LCH-based oil production processes presented in this study. Furthermore, vertical fermentation setups are seasonally independent compared to their agricultural counterparts, and in contrast to conventional agriculture, do not require fertilizers and pesticides [58]. Moreover, fermentative SCO production is not heavily affected by climate change and represents an area-efficient alternative to plant oils.

A main factor for the production cost of the SCO is the price of acetic acid, as it is one of the main carbon sources in this process. The high recent fluctuation in acetic acid prices can be explained by the limited number of suppliers, namely the USA and China, which supply the majority of the acetic acid, with the market shares of 17% and 55%, respectively [59]. Therefore, problems or even disruptions in the supply chains of these two countries have a profound impact on the pricing as the availability can be drastically decreased [44,60]. Local or even on-site production and supply of acetic acid would be the best route to avoid such dependencies and price fluctuations. In that context, up to 5% of wood weight is acetate esters, which could potentially be separated and used directly from pulp and paper waste streams as an acetic acid source [27,61].

In Europe alone, the pulp and paper industry produces around 11 million tons of waste every year [24]. Wood is mainly composed of 40–44% cellulose, 18–32% lignin, and 15–35% hemicelluloses [27,61]. The hemicellulose consists of xylan with 10–35% and acetate esters with 3–5% of the total wood [62,63]. Extrapolating these numbers to the actual biomass volume results in waste material of 85–151, 47–151, and 14–24 million metric tons of lignin, xylan, and acetate, respectively. Even a small proportion of these waste amounts would be enough to supply a biotechnological plant, as described here.

In the TEA, our lab-scale experiments for the biotechnological valorization of an industrial waste stream using *C. oleaginosus* were validated in an in silico industrial fermentation plant model. It was performed with a focus on the comparison of different conditions and raw material costs. Compared to glucose, the cost efficiency of LCH was demonstrated for the production of SCO on an industrially relevant scale. SCO can be used as a starting material for similar applications to those of plant-based oils in the biofuel sector as well as for the production of polymeric materials, lubricants, and other oleochemicals [64,65]. Such resource-efficient approaches could lead to more independent, circular, and sustainable economies in the future.

## 5. Conclusions

Single cell oils are a promising alternative to plant-based oils for the production of sustainable biofuels. This study provides a novel strategy to produce SCOs with the oleaginous yeast *C. oleaginosus*, utilizing lignocellulosic hydrolysate, a biogenic waste stream from the pulp and paper industry. To this end, fermentations were performed, and a maximal lipid titer of 42.1 ± 1.7 g/L (75.5% lipid per biomass) and a carbon conversion to lipid of 0.336 g/g were reached with optimized conditions. Furthermore, feeding strategies were tested to enhance the share of LCH in the total amount of fed carbon, where acetic acid has the main share. The most effective feeding strategy utilized a mix of acetic acid and LCH as the main carbon sources, both of which account for the main part of the waste stream from the pulp and paper industry. In summary, our data indicate that the xylose-rich LCH is a highly suitable substrate for the efficient bioconversion to SCO. By adding continuous feeding to the established consumption-based approach, the LCH share of consumed carbon was increased. The economic advantages of the use of LCH over glucose were shown in a techno-economic analysis. The predicted prices for the SCO produced moved in a range (2900 USD/t) that was competitive with sustainably produced organic plant oils (1000–3000 USD/t) [55,57]. At the same time, vertical fermentation

setups can be run in a seasonally independent manner, without the use of fertilizers and pesticides, and offer options for land-efficient scale-up. Our further research will continue in the direction of maximizing the reuse of industrial waste streams in an economically feasible way to close the remaining gaps for a circular bioeconomy and to contribute to the environmental challenges the world faces today. In conclusion, we present a sustainable and economical strategy to produce SCOs as an alternative platform to produce advanced biofuels, lubricants, and other oleochemical products.

## 6. Patents

Thomas B. Brueck, Mahmoud Masri, Nikolaus I. Stellner, and Zora S. Rerop have a European patent field with the Technical University of Munich involving the methodology described in this work.

**Supplementary Materials:** The following supporting information can be downloaded at: https://www.mdpi.com/article/10.3390/fermentation9020189/s1, Table S1: composition of the LCH, Table S2: pretreatment strategies, Table S3: sugar and organic acid composition of LCH before and after pretreatment, Figure S1: comparison between three different starting concentration of LCH, Table S4: Duncan groups of the statistical analysis of the fatty acid profile, Figure S2: GC-MS chromatograms of fermentation medium before and after fermentation, Table S5: share of LCH from the overall consumed carbon for the different feeding strategies, Figure S3: comparison of all five different feeding strategies as well as respective carbon conversion rates, Table S6: GC-FID relative quantification of LCH-derived lignin breakdown products before and after the fermentation, Figure S4: Super Pro Designer model for the TEA of SCO production on the basis of LCH and acetic acid.

**Author Contributions:** Conceptualization, T.B.B., M.M., N.I.S. and Z.S.R.; methodology, M.H., P.G., M.M., N.I.S. and Z.S.R.; software, M.M., N.I.S. and Z.S.R.; validation, T.B.B., M.M. and N.M.; investigation, P.G., M.M., N.I.S. and Z.S.R.; resources, T.B.B.; data curation, P.G., M.M., N.I.S. and Z.S.R.; writing—original draft preparation, N.I.S. and Z.S.R.; writing—review and editing, T.B.B. and M.M.; visualization, N.I.S. and Z.S.R.; supervision, T.B.B., N.M. and M.M.; project administration, M.M.; funding acquisition, T.B.B. All authors have read and agreed to the published version of the manuscript.

**Funding:** This research was funded by: The German Federal Ministry of Education and Research (Bundesministerium für Bildung und Forschung), grant number: FKZ 031B0662B; European Union's Horizon Europe research and innovation program, grant number: 101059786; Werner Siemens Foundation.

**Institutional Review Board Statement:** Not applicable.

**Informed Consent Statement:** Not applicable.

**Data Availability Statement:** The data presented in this study are available on request from the corresponding author. The data are not publicly available due to privacy protection.

**Acknowledgments:** We would like to thank Jeremias Widmann for his outstandingly diligent and dependable help with the sample preparation and analysis throughout our work. We also very much appreciate the expert knowledge and advice of Uwe Arnold, managing director of AHP Group GmbH, which led to the optimization of our techno-economic analysis setup. Finally, we express our thanks to Gülnaz Celik for her reliable support and advice in the fermentations performed for this study.

**Conflicts of Interest:** Thomas B. Brueck and Mahmoud Masri are board members at Global Sustainable Transformation GmbH.

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
