# Peer review of "Bioconversion of a Lignocellulosic Hydrolysate to Single Cell Oil for Biofuel Production in a Cost-Efficient Fermentation Process"

_fermentation, doi:10.3390/fermentation9020189_

Round 1
Reviewer 1 Report
Title:
Bioconversion of a Lignocellulosic Hydrolysate to Single Cell Oil for Biofuel Production in a Cost-Efficient Fermentation Process
Waste-to-Energy is widely discussed nowadays to reduce the negative impacts of using fossil fuels and to ensure a sustainable energy supply. Yeast have been discussed as promising feedstock for energy production, coupled with waste recycling. The present study examined the metabolic process and utilization of industrial paper mill lignocellulosic hydrolysate for cultivation of Cutaneotrichosporon oleaginosus. Overall, the study is of interest and within the scope of Fermentation. However, I have the following comments;
1. The authors should clarify that the used microorganism belongs to the yeast and add the identification author’s name for that yeast’ first mention in the manuscript.
2. Line#128, please clarify what type of water is used in washing.
3. Line# 136, replace Italic F with normal F in “For”
4. Please clarify Lines# 150-152 “The fatty acid profile was measured through gas chromatography after fatty acid methyl esterification (FAME) of unwashed and lyophilized samples from the fermentation process.”
5. Line# 175, change CO2 to CO2
6. Line# 176, be constant of using units rpm or rcf and minutes or min.
7. Line# 185, replace “(0.05 g/L kanamycin, 0.1 g/L ampicillin)” with (0.05 g/L kanamycin and 0.1 g/L ampicillin)
8. Line# 210 and 214, do the authors mean by pH change pH monitoring?
9. There is no statistical analysis throughout the manuscript. I recommended adding the Duncan test in all figures.
10. There is no title for the second y-axis in figure 2, S1 or 3.
11. Lines# 320-322: the authors mention the growth rate as g/L/h but this considers biomass productivity not the growth rate. Moreover, it’s not mentioned in figure S1 as mentioned by the authors. As well as there is no any data about lipid productivity in the manuscript as mention in the material and methods.
12. the conversion yields in line 379 are not plotted in fig. 3, as mentioned by the authors.
13. There are many of data in the result that didn’t Plot or mentioned in any figure or table.
14. there is no figure or table cited in section 3.8. Fermentation with co-feeding of acetic acid and lignocellulosic hydrolysate
15. XGA is not plotted in Fig.2 a, as mentioned by the authors.
16. please correct the acetic acid: LCH ratio in section 3.7. Screening of different strategies for feeding lignocellulosic hydrolysate sometimes wrote 50:00 and others 50:10, and it should be 50:100.
17. Discussion is well presented but lacking discussion about the fatty acid profile and is it suitable for biodiesel production, especially the unsaturation fatty acid level of the studied yeast is higher than the saturated one.
You can use such references:
https://doi.org/10.1016/j.algal.2020.102088
https://doi.org/10.1016/j.envres.2021.112100
Author Response
We thank the reviewer very much for the intensive and thoughtful revision. We appreciate your meaningful comments and improved our manuscript accordingly.
Point 1. The authors should clarify that the used microorganism belongs to the yeast and add the identification author’s name for that yeast’ first mention in the manuscript.
Response 1. Yeast specification and ATCC number were included in the introduction and the full name and IDs in the material and method section.
Point 2. Line#128, please clarify what type of water is used in washing.
Response 2. The term “ultrapure water” was added to specify the water characteristics.
Point 3. Line# 136, replace Italic F with normal F in “For”
Response 3. The formatting mistake was corrected.
Point 4. Please clarify Lines# 150-152 “The fatty acid profile was measured through gas chromatography after fatty acid methyl esterification (FAME) of unwashed and lyophilized samples from the fermentation process.”
Response 4. We agree with the reviewer, the sentence was split, to clarify the meaning and an explanation was added.
Point 5. Line# 175, change CO2 to CO2
Response 5. The formatting issue was corrected.
Point 6. Line# 176, be constant of using units rpm or rcf and minutes or min.
Response 6. The inconstant use of rpm and minutes instead of rcf and min in this section was corrected.
Point 7. Line# 185, replace “(0.05 g/L kanamycin, 0.1 g/L ampicillin)” with (0.05 g/L kanamycin and 0.1 g/L ampicillin)
Response 7. “and” was included in the parenthesis.
Point 8. Line# 210 and 214, do the authors mean by pH change pH monitoring?
Response 8. “…, a feed of 90% acetic acid solution used for pH regulation. At the same time, acetic acid functioned as feed for the cultivation.” Was added to explain the feeding system that was used.
Point 9. There is no statistical analysis throughout the manuscript. I recommended adding the Duncan test in all figures.
Response 9. Thank you for the important point. Duncan test were added for lipid titers and carbon conversion within figures 2a and 2b. For the fatty acid profile (Fig. 2c) a table with Duncan groups for all relevant fatty acids was added. Further, a short explanation was added to the material and methods section. For other figures, no statistical indications were added, as the comparison would make the figure very complex to understand. The important analysis of the lipid titer, carbon conversion, and fatty acid profile is shown, as mentioned in Fig. 2. If you would recommend including Duncan test in the other figures, please let us know.
Point 10. There is no title for the second y-axis in figure 2, S1 or 3.
Response 10. The axis titles were added. However, the scale is the same for all values displayed, except for Figure 1a, where the scale was adjusted due to readability.
Point 11. Lines# 320-322: the authors mention the growth rate as g/L/h but this considers biomass productivity not the growth rate. Moreover, it’s not mentioned in figure S1 as mentioned by the authors. As well as there is no any data about lipid productivity in the manuscript as mention in the material and methods.
Response 11. Thank you for pointing us to this, the unit was wrongly added, as it is the growth-rate coefficient resulting from fitting with Gompertz function. Further a small addition about Gompertz usage was made in the material and method section. To avoid misunderstandings the reference for the supplementary figure S1 was moved, as it should point to the fermentation overview (including Gompertz fitted growth curves), but not refer to explicit function tables. If further data on the Gompertz parameters is required, we are happy to provide it.
Lipid conversion yield was calculated as mentioned in the material and methods section and further used for the TEA, but as you pointed out, the values were not included in the text. We included them now in sections 3.6. and 3.8.
Point 12. the conversion yields in line 379 are not plotted in fig. 3, as mentioned by the authors.
Response 12. The plot of conversion yields was included in the supplementary material and referenced.
Point 13. There are many of data in the result that didn’t Plot or mentioned in any figure or table.
Response 13. The two missing conditions were included in the supplementary material.
Point 14. there is no figure or table cited in section 3.8. Fermentation with co-feeding of acetic acid and lignocellulosic hydrolysate
Response 14. The figure for the combined co-feeding strategy is figure 1e, which was not mentioned in the text. We included the reference.
Point 15. XGA is not plotted in Fig.2 a, as mentioned by the authors.
Response 15. Figure 2 was adjusted to include XGA as an additional control setting.
Point 16. please correct the acetic acid: LCH ratio in section 3.7. Screening of different strategies for feeding lignocellulosic hydrolysate sometimes wrote 50:00 and others 50:10, and it should be 50:100.
Response 16. It is supposed to be 50:10, as it means that the feed consists of 50% acetic acid and 10% LCH, and the rest is ultrapure water. We added a detailed description in parenthesis.
Point 17. Discussion is well presented but lacking discussion about the fatty acid profile and is it suitable for biodiesel production, especially the unsaturation fatty acid level of the studied yeast is higher than the saturated one.
You can use such references:
https://doi.org/10.1016/j.algal.2020.102088
https://doi.org/10.1016/j.envres.2021.112100
Response 17. Thank you for the vital point, a section about the application of the produced SCO as biofuel was added to the discussion and the suggested references were included.
Reviewer 2 Report
A process to produce yeast oil, using a new single-cell organism, C.oleaginosus, by fermentation of a waste from paper manufacture (lignocellulosic hydrolysate) is studied and developed.
The Abstract is concise and at the same time descriptive. The Introduction could stand as a mini-review, proving a good documentation from up-to-date literature and good structure of ideas. Section 2 Materials and methods is ample and gives the details needed to understand and replicate the experiment, and a good description of the techno-economic analysis as a model for other works. Results are provided extensively in the Supplementary materials, and concentrated in the article’s figures and in the text. Discussion contains clear explanation of the results, compared with other works on the topic. Conclusions are pertinent and stress on the major achievements of the work.
Small editing mistakes to be corrected:
-p.3, line 140: “after modifying Bligh and Dyer method” instead of “after modifying Bligh and Dyer”
-p.12, line 471: “form” instead of “from”
-p.17,line 708: “[41] A.A.D.Humbird” instead of “[41] and A.A.D.Humbird”
Author Response
We thank the reviewer very much for the constructive and positive feedback. We have adressed the mentioned points to improve our manuscript accordingly.
Point 1. p.3, line 140: “after modifying Bligh and Dyer method” instead of “after modifying Bligh and Dyer”
Response 1. The inaccuracy was corrected accordingly.
Point 2. p.12, line 471: “form” instead of “from”
Response 2. The spelling mistake was corrected accordingly.
Point 3. p.17,line 708: “[41] A.A.D.Humbird” instead of “[41] and A.A.D.Humbird”
Response 3. The incorrect information was adjusted in the citation program and the reference list was refreshed and is correct now.
Reviewer 3 Report
See the attached file.

Author Response
We thank the reviewer very much for the intensive and thoughtful review. We appreciate your constructive comments and have improved our manuscript accordingly.
Point 1. Abstract: ...which in addition to hexoses and pentoses can metabolize organic acids.
Response 1. The sentence is corrected accordingly.
Point 2. Introduction (line 70): “The total production of pulp and paper was 188.9 million metric
tons in 2020”. Authors should mention if this value corresponding to the global production
or other geographic area.
Response 2. The numbers refer to a global scale, this information was added to the text.
Point 3. Please avoid results (such as “resulting in high lipid titers”) from the introduction section.
Response 3. The inadvertent result mentioned in the introduction section was removed.
Point 4. The materials and methods section can be organized with major categories such as raw
material analysis, strain and media, fermentation, analytical studies, and technoeconomic
analysis. This section need to be revised with a logical order considering the research
design. Generally, the materials and methods section was presented arbitrarily.
Response 4. The material and method section was re-structured into the subsections: Analysis of lignocellulosic hydrolysate, Strain and media composition, Fermentation, Monitoring of fermentation, and Technoeconomic analysis. It is now much better organized.
Point 5. Authors should mention the details about the collection of LCH to conduct this study.
Response 5. The LCH was provided by an industrial paper mill. We have a confidentiality agreement that does not allow us to specify any further data about its origin.
Point 6. There is no clear demarcation between the raw LCH and the pretreated ones, including if
both the raw and pretreated LCH were analyzed, used for fermentation, etc. Section 3.2
presented the results of pretreated LCH. Authors didn’t discuss the alternation in the
chemical composition of the pretreated LCH including the sugar content and others.
Response 6. An additional table S3 was added to the supplementary material to display the sugar and organic acid composition before and after the pre-treatment of the LCH. As now mentioned in section 3.2. and also in the material and methods section 2.2.2., all the fermentation experiments were conducted with the pretreated LCH.
Point 7. 2.12 should be ‘Bioreactor fermentation’.
Response 7. The heading was corrected accordingly.
Point 8. The bioreactor size (volume) should be indicated.
Response 8. The total reactor volume was added in the material and method section.
Point 9. Figures should be arranged in a manner to be cited in the manuscript with a proper order.
E.g., line 289, Fig. 1f.
Response 9. Figures were adjusted and re-checked in the text. We hope, that everything is in a stringed order now.
Point 10. Some typos need to be double checked (e.g., line 535, 557...).
Response 10. The two mentioned typing errors were corrected and a last check for others was done as well.
Point 11. Please use SCO consistently after ‘single cell oil’ is abbreviated in its first appearance. This
applies for all abbreviations.
Response 11. Nearly all not abbreviated single cell oil, lignocellulosic hydrolysate, and techno-economic analysis were replaced with the abbreviations. Exceptions were made for the abstract, subtitles, material and methods (first mention), explanations in figure captions, and the first mention in the conclusion, as these parts may stand alone and should be understood without further explanation.
Point 12. Use uniform format for list of references, particularly the position of publication year.
Response 12. The reference list was created with the citation software Mendeley and should be concise. Online references are cited with the publication year in the middle, all other references should have the year at the end. In two cases the year was not correctly placed, so we double-checked the library and corrected these cases.
Round 2
Reviewer 3 Report
Authors have addressed most of the comments.